# A Comprehensive Computational Investigation into the Conserved Virulent Proteins of *Shigella* species Unveils Potential Small-Interfering RNA Candidates as a New Therapeutic Strategy against Shigellosis

**DOI:** 10.3390/molecules27061936

**Published:** 2022-03-17

**Authors:** Parag Palit, Farhana Tasnim Chowdhury, Namrata Baruah, Bonoshree Sarkar, Sadia Noor Mou, Mehnaz Kamal, Towfida Jahan Siddiqua, Zannatun Noor, Tahmeed Ahmed

**Affiliations:** 1International Centre for Diarrhoeal Disease Research, Bangladesh (icddr,b), Dhaka 1212, Bangladesh; parag.palit@icddrb.org (P.P.); mehnaz.kamal@icddrb.org (M.K.); towfida@icddrb.org (T.J.S.); tahmeed@icddrb.org (T.A.); 2Department of Biochemistry and Molecular Biology, University of Dhaka, Dhaka 1000, Bangladesh; farhanatasnim@du.ac.bd (F.T.C.); bonoshreesarkar98@gmail.com (B.S.); mousadia5@gmail.com (S.N.M.); 3Department of Biological Sciences and Bioengineering, Indian Institute of Technology, Kanpur 208016, Uttar Pradesh, India; nmrtab@gmail.com

**Keywords:** *Shigella*, small-interfering RNA, computational algorithms, RNAi pathway, shigellosis

## Abstract

*Shigella* species account for the second-leading cause of deaths due to diarrheal diseases among children of less than 5 years of age. The emergence of multi-drug-resistant *Shigella* isolates and the lack of availability of *Shigella* vaccines have led to the pertinence in the efforts made for the development of new therapeutic strategies against shigellosis. Consequently, designing small-interfering RNA (siRNA) candidates against such infectious agents represents a novel approach to propose new therapeutic candidates to curb the rampant rise of anti-microbial resistance in such pathogens. In this study, we analyzed 264 conserved sequences from 15 different conserved virulence genes of *Shigella* sp., through extensive rational validation using a plethora of first-generation and second-generation computational algorithms for siRNA designing. Fifty-eight siRNA candidates were obtained by using the first-generation algorithms, out of which only 38 siRNA candidates complied with the second-generation rules of siRNA designing. Further computational validation showed that 16 siRNA candidates were found to have a substantial functional efficiency, out of which 11 siRNA candidates were found to be non-immunogenic. Finally, three siRNA candidates exhibited a sterically feasible three-dimensional structure as exhibited by parameters of nucleic acid geometry such as: the probability of wrong sugar puckers, bad backbone confirmations, bad bonds, and bad angles being within the accepted threshold for stable tertiary structure. Although the findings of our study require further wet-lab validation and optimization for therapeutic use in the treatment of shigellosis, the computationally validated siRNA candidates are expected to suppress the expression of the virulence genes, namely: IpgD (siRNA 9) and OspB (siRNA 15 and siRNA 17) and thus act as a prospective tool in the RNA interference (RNAi) pathway. However, the findings of our study require further wet-lab validation and optimization for regular therapeutic use for treatment of shigellosis.

## 1. Introduction

Shigellosis can be attributable to approximately 12.5% cases of diarrheal mortality on a global scale; accounting for about 163,400 cases of annual deaths, 54,900 of these cases are children of less than five years of age [1]. The global burden of shigellosis mainly involves *S. flexneri* and *S. sonnei*, whereas *S. boydii* is uncommon outside South-east Asia [2]. In regions with rapid industrialization and development in water sanitation and in economically developing regions, *S. sonnei* has gained prevalence in the epidemiological shift in places like Vietnam, Thailand, and Bangladesh [3,4,5,6]. *S. sonnei* is also predominant in traveler’s diarrhea [7].

Although antibiotic-resistant strains are being reported against ciprofloxacin, azithromycin, and ceftriaxone, these continue to be utilized as mainstay treatment strategies [8]. The growing rise of antimicrobial resistance to *Shigella* sp. has thus led to the urgent need for development of newer therapeutic strategies against shigellosis [9,10,11,12,13,14,15]. A multi-centric study reported that 85% of cases occurring in LMIC could be attributable to *S. flexneri* 2a, 3a, and 6 together with *S. sonnei*, and, therefore, a quadrivalent vaccine targeting these strains is expected to provide significant protection in endemic regions [16]. Serotype-based vaccines include conjugate vaccine, carbohydrate vaccine, and live-attenuated or killed whole-cell vaccines [17,18,19,20]. However, due to the lack of ideal animal models or low/serotype-specific protection, no *Shigella* vaccine has reached the stage of commercialization to date [21].

The concept of designing a single therapeutic candidate with substantial efficacy and potency against all *Shigella* sp. would thereby involve targeting multiple conserved proteins involved in the process of invasion and pathogenesis. The virulence-related proteins or Vir proteins of *Shigella* such as VirA help in entry and intracellular motility and golgi fragmentation in the host [22,23]; VirB activates the invasion proteins [24], and VirF, is involved in cell invasion and activation of VirG and VirB [25]. VirG, also called IcsA, helps in actin polymerization and hence the movement of the bacteria from one cell to another [26]. The invasion plasmid antigens, Ipa, are the key regulators of invasion and pathogenicity and also activate other related proteins [27,28,29,30]. IpgB is an invasion effector protein of the Type III Secretion system (T3SS), involved in membrane ruffling for cellular entry of *Shigella* [31]. IpgD is another effector involved in entry and survival of the bacteria [32,33,34]. There are many membrane excretion proteins and surface presentation antigens together called Mxi-Spa proteins that make up the T3SS and help in pathogenesis and invasion. In particular, Spa 33 belonging to the class of gatekeeper proteins controls T3SS protein secretion into host cells and helps in the secretion of Ipa proteins [35], while MxiC interacts with the Ipa proteins to mediate entry and the subsequent survival of the bacteria [36]. Other T3SS effectors such as OspB, OspF, and OspG modulate host cell-signaling pathways and transcription and manipulate the host inflammatory response [34,37,38].

Recent developments in the quest for newer therapeutic strategies against a number of infectious pathogens have involved the utilization of the concept of post-transcriptional gene silencing by designing siRNA molecules specific to the pathogen [39]. Small-interfering RNA (siRNA) represent a class of exogenous double-stranded non-coding RNA that bind to specific sequences of the related messenger RNA (mRNA) and promote the process of degradation of the mRNA, eventually halting the process of transcription [40]. Concurrent works involving the use of a plethora of computational tools and algorithms have designed potential therapeutic siRNA candidates against a number of infectious organisms, including *P.vivax* [41] and *L.donovani* [42]; the COVID-19 pandemic causing SARS-CoV-2 [43]; and against a number of bacterial pathogens, such as: *M. tuberculosis* [44], *Salmonella* [45], and *Listeria* [46].

In this study, we aimed to design novel therapeutic options against all members of the *Shigella* genus that are pathogenic to humans (*S. flexneri, S. dysenteriae, S. sonnei*, and *S. boydii*) through the design of siRNA candidates against a number of conserved *Shigella* proteins, which are involved in the process of invasion and pathogenesis. We targeted a total of 15 conserved virulent proteins of *Shigella* sp., including: IcsA/Vir G, IpaA, IpaB, IpaC, IpaJ, IpgB, IpgD, MxiC, OspB, OspF, OspG, Spa33, VirA, VirB, and Vir F using rigorous computational tools and algorithms for the selection of the most feasible and effective siRNA candidates against *Shigella*.

## 2. Methods and Materials

An overview of the methodology followed in our study is illustrated in Figure 1. A complete list of all the webservers used in this study is given in Table 1.

### 2.1. Sequence Retrieval and Multiple Sequence Alignment for Determination of Conserved Regions

Complete gene sequences for 15 different conserved virulent proteins of multiple isolates of *Shigella* sp. were obtained from NCBI Nucleotide (https://www.ncbi.nlm.nih.gov/nucleotide) [47]. These conserved virulent proteins of *Shigella* sp. included: IcsA (from 40 isolates), IpaA (from 64 isolates), IpaB (from 55 isolates), IpaC (from 41 isolates), IpaJ (from 57 isolates), IpgB (from 46 isolates), IpgD (from 42 isolates), MxiC (from 62 isolates), OspB (from 68 isolates), OspF (from 28 isolates), OspG (57 isolates), Spa33 (from 21 isolates), VirA (from 50 isolates), VirB (from 61 isolates), and VirF (68 isolates). Identification of conserved regions from these 15 different virulent proteins of *Shigella* sp. was performed by multiple sequence alignment using Clustal Omega (https://www.ebi.ac.uk/Tools/msa/clustalo/) [48].

### 2.2. Recognition of Target Sequence and Designing of Potential siRNA Candidates

For the purpose of identification of target sequence and siRNA designing, siDirect 2.0 (http://sidirect2.rnai.jp/), an efficient and target-specific siRNA designing tool was used [49]. This tool employs a combination of first-generation algorithms for siRNA designing, namely: Ui-Tei, Amarzguioui, and Reynolds rules (URA rules) along with a melting temperature of less than 21.5 °C as the absolute parameters for prediction of potential siRNA duplex formation [50]. The components of the URA rules as shown in Appendix A have been used in a number of previously published studies [42].

The i-SCORE Designer, an online software [51], was used for the validation of the potential siRNA candidates obtained from siDirect 2.0. This web-based software utilizes multiple second-generation algorithms in addition to the first-generation algorithms for siRNA designing (U, R, and A rules), among which i-SCORE, s-Biopredsi, Katoh, and DSIR rules are the prominent second-generation algorithms [52,53,54]. Only those siRNA candidates that were found to fulfill the threshold set by the criteria for the second-generation algorithms were subsequently selected for further downstream validation.

### 2.3. Determination of Off-Target Similarity

The off-target sequence similarity for the guide strands of the siRNA candidates was checked using the BLAST tool (http://www.ncbi.nlm.nih.gov/blast), accessed on 12 January 2021 [55] against the entire GenBank database with default threshold e-value of 0 and BLOSUM 62 as a parameter.

### 2.4. Prediction of Free Energy of Folding and Calculation of GC Content

The free energy of folding of the guide strand of the siRNA candidates was assessed using RNAstructure (https://rna.urmc.rochester.edu/RNAstructureWeb/) [56], a web-based tool for prediction of the secondary structure of RNA [56]. Only those siRNA candidates that had exhibited a positive free energy of folding (positive ∆G) were used for the subsequent process of GC content calculation by using OligoCalC: Oligonucleotide Properties Calculator (http://biotools.nubic.northwestern.edu/OligoCalc.html) [57] and siRNA candidates with a GC content of 30–60% that were subsequently selected for downstream validation.

### 2.5. Evaluation of the Thermodynamics Involved in the Formation of Secondary Structure of siRNA and Target and Visualization of siRNA-Target Binding

Investigation of the thermodynamics for the secondary structure formed between the guide strand and subsequent validation of the siRNA-target duplex was performed using the web-based tool for RNA secondary structure prediction, RNAstructure (https://rna.urmc.rochester.edu/RNAstructureWeb/) [56]. This tool determines the hybridization energy and base pairing from two RNA sequences by following the functional algorithm of McCaskill’s partition to compute probabilities of base pairing, realistic communication energies, and equilibrium concentrations of duplex structures [56,57,58].

### 2.6. Determination of Heat Stability and Prediction of Functional Efficiency and Target Accessibility of the Potential siRNA Candidates

For the determination of heat stability of the guide strand as well as for the assessment of functional efficiency and target accessibility of the predicted siRNA candidates, OligoWalk (http://rna.urmc.rochester.edu/cgi-bin/server_exe/oligowalk/oligowalk_form.cgi), a web-based server for calculating thermodynamic features of sense-antisense hybridization was used [59]. This tool operates through a designated query involving the sequence for the guide strand of the siRNA candidate, and the subsequent results obtained are expressed as: “End-diff (free energy difference between the 5′ and 3′ end of the antisense strand of siRNA)”, which indicates the functional efficiency of the siRNA candidate; “Break-targ. ∆G (free energy cost for opening base pairs in the region of complementarity to the target)”, which signifies the target accessibility of the designed siRNA; and “Probability score of being efficient siRNA”, which is calculated on the basis of both target accessibility and functional efficiency of the designed siRNA [59,60].

### 2.7. Validation of the Functional Efficiency of the siRNA Candidates

The siRNAPred server (http://crdd.osdd.net/raghava/sirnapred/index.html) was used for the validation of the functional efficiency of the predicted siRNA candidates against the Main21 dataset using a support vector machine algorithm and the binary pattern-prediction approach [61]. Validation scores from the server greater than 1 predict very high efficiency; scores ranging from 0.8–0.9 predict high efficiency; and scores ranging from 0.7–0.8 predict moderate efficiency [61].

### 2.8. Prediction of Immunotoxicity of the Predicted siRNA Candidates

For the purpose of predicting the immunotoxicity of the designed siRNA candidates, imRNA (https://webs.iiitd.edu.in/raghava/imrna/sirna.php) (accessed on 12 January 2021), a web-based server consisting of various components integral for the designing of RNA-based therapeutics, was employed [62]. An IMscore of 4.5 set as the default was used for the prediction of immunogenicity of potential siRNA candidates.

### 2.9. Designing of Tertiary (3D) Structure of the siRNA Candidates and Validation

Designed siRNA candidates that had surpassed the immunogenicity filter set in the previous step were subjected to tertiary structure prediction using RNAComposer (http://rnacomposer.cs.put.poznan.pl/) (accessed on 12 January 2021), an automated tool that predicts the tertiary structure from a linear siRNA sequence [63,64]. Subsequently, the tertiary structures of these siRNA candidates were validated using the MOLprobity web server (http://molprobity.biochem.duke.edu/) [65], which uses the pdb file of the tertiary structure of the siRNA as input to predict the validity of the tertiary structure on the basis of all-atom contacts and geometry, RNA sugar puckers, RNA backbone conformations, hydrogen bonds, and Van der Waals forces [65,66]. The tertiary structures of the candidate siRNAs with an acceptable tertiary structure were visualized using the PyMOL Molecular Graphics System (v1.8.4).

## 3. Results

### 3.1. Retrieval of Nucleotide Sequences of Conserved Virulent Proteins of Shigella sp.

A total of 264 conserved sequences (shown in Appendix A) from 15 different conserved virulent *Shigella* proteins were obtained from subsequent multiple-sequence-alignment analysis performed using Clustal Omega.

### 3.2. Designing siRNA Candidates by Using a Combination of First-Generation and Second-Generation Algorithms

siDirect 2.0, a highly efficient web-based computational tool, was used to predict potential target-specific siRNA candidates on the basis of three first-generation algorithms governing the sequence preference of siRNA, namely: Ui-Tei, Amarzguioui, and Reynolds rules (U, R, and A rules) [49]. This tool predicted a total of 58 potential siRNA candidates from 264 conserved regions of the conserved virulent *Shigella* proteins, which were found to comply with the first-generation algorithms for siRNA designing, i.e., -U, R, and A rules.

Out of these 58 siRNA candidates that fulfilled the U, R, and A rules, only 38 siRNA candidates were found to comply with the 90% threshold set by the i-SCORE Designer software for the second-generation algorithms for siRNA designing [51]. Table 2 illustrates the results of the analysis obtained from the first- and second-generation algorithms of siRNA designing. The NCBI BLAST program was used to confirm off-target resemblance of siRNAs, and none of the 38 siRNA candidates that had fulfilled both the first- and second-generation algorithms for siRNA designing had displayed any off-target similarity with the human genome.

The function of siRNA largely depends on the molecular structure. Towards this, extensive efforts have been undertaken in predicting the secondary structure of the RNA molecules [67]. The benchmark of the molecular structural accuracy of the siRNA was set as the “Minimum Free Energy (MFE)” [68]. The minimization of free energy is an established phenomenon in computational structural biology on the principle that at a state of equilibrium, the molecule folds into the state of least energy [69]. The minimum free energy of folding was calculated by using the RNAstructure web server to assess the stability of the predicted siRNA guide strands. From the above-mentioned analysis, 30 siRNA candidates were found to yield a positive free energy of folding and thus were considered for further analysis (Table 2).

The GC content of siRNA is a major determinant of the stability of the secondary structure of the siRNA, whereby a GC content ranging from 30 to 52% is considered sufficient for the execution of its action [70]. In our study, out of the 30 siRNA candidates that exhibited a positive free energy of folding in addition to complying with all the primary and secondary algorithms, only 19 candidates were found to have a GC content in between 30–52% and thus were selected for subsequent analysis (Table 2).

### 3.3. Evaluation of Binding Energy and Visualization of Secondary Structures of siRNA-Target Duplex

Precise prediction of binding energy between siRNA and the target is integral for a proper understanding of the binding of siRNA to the target and for the subsequent assessment of functional efficiency of the siRNA [71,72,73]. The binding energy of siRNA with the target is a predictive score to account for the energy cost of opening up the nucleotides in the mRNA strand to allow hybridization to siRNA, so that all the nucleotides in the hybridization site are forced into a single-stranded conformation [74]. The prediction of binding energy of siRNA to the target also assumes that siRNA binding results in the re-equilibration of the complete target secondary structure [74].

RNAstructure, an online web server, was used for the estimation of the hybridization energy between the siRNA–target duplex. The thermodynamics of the siRNA–target interaction and the details of the calculation of this binding energy between siRNA and the target have been published elsewhere [75,76]. Table 3 shows the binding energies of the siRNA with the target sequence.

Secondary structures of siRNA with their respective targets provide an efficient computational estimation for both the structure and thermodynamics of RNA–RNA interaction [71]. The RNAstructure program predicts the most stable secondary structures of the target–siRNA duplex and minimizes the folding energy. The temperature chosen to predict the folded structure was 37 °C. The secondary duplex of candidate siRNA molecules and their corresponding targets are elucidated in Figure 2.

### 3.4. Determination of Heat Stability of siRNA–Target Duplex

Heat stability analysis of the siRNA–target duplex, a key factor in the assessment of the stability of secondary structure and the functional efficiency of siRNA [77], was conducted by using the OligoWalk server. The results of the heat-stability analysis of the siRNA–target duplex showed that all the potential 19 siRNA candidates that had fulfilled all the criteria for effective siRNA assessed so far (as shown in Table 1) had a melting temperature of greater than 70 °C (Table 3). Henceforth, the melting temperatures of each of the siRNA–target duplex structures were found to be substantially greater than the physiological temperature of 37.4 °C indicating towards the maintenance of the integrity of the secondary structure of siRNA in the host physiological system.

### 3.5. Prediction of Functional Efficiency and Target Accessibility of the siRNA Candidates

Table 2 delineates a summary of the results of the analysis of the functional efficiency and target accessibility as determined by the OligoWalk web server, of each of the siRNA candidates that had fulfilled the threshold for all the criteria set in Table 2. In Table 3, the “End-diff” score for each of the 19 potential siRNA candidates indicates the free energy difference between the 5′ and 3′ end of the antisense strand of siRNA [60]; the siRNA candidates that have positive “End-diff” scores were ranked to have a high functional efficiency. In our study, all the siRNA candidates demonstrated a positive ‘End-diff’ score, thus indicating towards a high functional efficiency of the candidate siRNAs. Moreover, siRNA 9, siRNA 13, siRNA 39, siRNA 15, and siRNA 23 were found to exhibit more positive “End-diff” scores compared to the rest of the siRNA candidates and thus can be classified as having the highest functional efficiency. Consequently, target accessibility was determined by the “Break targ. ΔG” score, which is a free energy account for the opening of base pairs in the region of complementarity to the target [77]. Candidate siRNAs demonstrating a less-negative “Break targ. ΔG” score are classified as having greater target accessibility [60,61,62,63,64,65,66,67,68,69,70,71,72,73,74,75,76,77]. In our study, siRNA 5, siRNA 6, siRNA 49, and siRNA 50 showed the least-negative “Break targ. ΔG” score and thus can be classified as having the greatest target accessibility.

Another score delineated in Table 2 as “Probability of being efficient siRNA” manifests both the functional efficiency and target accessibility of the designed siRNA candidates [60,61,62,63,64,65,66,67,68,69,70]. In our analysis, siRNA 50, siRNA 51, and siRNA 54 demonstrated a “Probability of being efficient siRNA” score of less than 0.9 and were excluded from further analysis. The rest of the siRNA candidates showed a “Probability of being efficient siRNA” score of greater than 0.9 and thus could be predicted to have high functional efficiency and substantial target accessibility.

### 3.6. Validation of the Functional Efficiency of the siRNA Candidates

Among the 16 predicted siRNA candidates that had fulfilled the threshold for all the criteria set so far, 11 siRNA candidates (siRNA 4, siRNA 5, siRNA 6, siRNA 9, siRNA 13, siRNA 23, siRNA 39, siRNA 42, siRNA 46, siRNA 53, and siRNA 57) showed a siRNA validity score between 0.8–1.0, following the binary pattern-prediction approach indicating high functional efficiency [61]. On the other hand, five siRNA candidates (siRNA 15, siRNA 17, siRNA 32, siRNA 38, and siRNA 49) showed a binary validation score of greater than 1.0, which manifests towards the highest functional efficiency [61].

### 3.7. Prediction of Immunotoxicity of the siRNA Candidates

We evaluated the immunotoxicity of the 16 siRNA candidates for which functional efficiency was validated in the previous step. Eleven out of the 16 siRNA candidates were found to be non-immunogenic on the basis of the default threshold IMscore set for assessing potential immunogenicity of query siRNA sequences and were considered for subsequent analysis. siRNA 23, siRNA 32, siRNA 42, siRNA 53, and siRNA 57 were found to be immunogenic (an IMscore greater than the threshold of 4.5) and were discarded.

### 3.8. Designing of Tertiary Structure of the siRNA Candidates and Validation

Respective tertiary/3D structures for each of the 11 non-immunogenic siRNA candidates were generated using the web-based RNA modeling software, RNAComposer [78], and the individual 3D structures were saved in pdb format for downstream application.

Subsequently, the MOLProbity web server was used to validate the individual tertiary structures of the non-immunogenic siRNA candidates, and the 3D structures were then filtered for clash score/the number of serious steric overlaps (>0.4 Å) per 1000 atoms [65]. In the MOLProbity web server, the threshold for the different aspects of nucleic acid geometry, including the probability of wrong sugar puckers, bad backbone confirmations, and bad angles was considered to be 5%, as recommended by the guidelines of the web server and by previously published research that had used this web server [41,65,79]. Moreover, the score for all atom contacts was considered to be acceptable if the clashscore was greater than or equal to the 33rd percentile, as recommended by previous literature [41,79]. Validation scores for the predicted 3D structures of the siRNA candidates are shown in Table 4. The acceptable tertiary structures of these siRNAs was based on algorithms designed to accommodate the X-ray crystallography models of these molecules, thereby implying that these siRNAs with an acceptable tertiary structure would be likely to retain both its structural feasibility and functional efficiency in both the physiological environment and in the aqueous solution.

In our study, a total of three siRNA candidates, namely: siRNA 9, siRNA 15, and siRNA 17, showed an acceptable tertiary structure with all the scores of the analyzed criteria for nucleic acid geometry such as: the probability of wrong sugar puckers, bad backbone confirmations, bad bonds, and bad angles being below the 5% threshold of acceptance for tertiary structures. Figure 3 illustrates the tertiary structures of siRNA 9, siRNA 15, and siRNA 17 as visualized using the PyMol Molecular Graphics System (v1.8.4).

## 4. Discussion

Our computational study is concurrent with the current efforts in the development of new therapeutic strategies to combat the persistent conundrum surrounding the emergence of multi-drug resistance among *Shigella* sp. To the best of our knowledge, this is the first study that has envisaged proposing a novel treatment option for shigellosis by targeting conserved virulent proteins of *Shigella*.

In our study, we designed siRNA candidates that are expected to target the expression of the specific conserved virulence genes of *Shigella* sp. through gene silencing [80]. Designing of siRNA candidates represents a new therapeutic strategy aimed to induce RNA specific inhibition [81], whereby an effective siRNA must fulfill all the threshold criteria set by the first-generation (Ui-Tei, Amarzguioui, and Reynolds rules) and second-generation (i-SCORE, s-Biopredsi, DSIR, and Katoh rules) algorithms for siRNA designing [82,83]. In this current study, for all the 15 aforementioned conserved virulence genes of *Shigella* sp., we obtained a total of 58 siRNA candidates that fulfilled the first-generation algorithms, among which only 38 siRNA (Table 2) candidates satisfied all the second-generational algorithms. Consequently, off-target silencing of siRNAs can potentially lead to undesired toxicities [84], and none of the siRNA candidates that satisfied both the first- and second-generation algorithms were found to exhibit such off-target activity within the human genome.

The folded secondary structure of siRNA is integral for assessment of functional efficiency of siRNA [67], and for the evaluation of structural stability and accuracy of RNA, minimum free energy (MFE) is considered as a standard parameter [68]. In our study, out of all the siRNA candidates satisfying both first- and second-generation algorithms, 30 siRNA candidates (Table 2) showed a positive MFE value, indicating the thermodynamic non-feasibility of self-folding. Subsequently, only 19 of these 30 siRNA candidates with a positive MFE value were found to have GC content within the range of 30–52%. GC content between 30% and 52% is recommended for the siRNA sequence, since there is a considerable reciprocal correlation between GC content and RNAi activity [70], whereby a GC content of siRNA within the aforementioned range exhibits stronger inhibitory effect [85].

The prediction of secondary structure between siRNA and target mRNA acts as an integral cue to the selection of a specific siRNA target site [85]. Random folding of siRNA may lead to the inhibition of its RNAi activity, and with an inappropriate secondary structure of siRNA-duplex, there can be an impediment to the RNA-induced-silencing-complex (RISC) formation [86]. Henceforth, evaluation of the thermodynamic outcome for the interaction between the siRNA candidate and the target mRNA involves the assessment of the sum of the energy required to unravel the binding site and the energy gained from the resultant hybridization process [75]. In our study, all the 19 siRNAs that were analyzed for feasibility in binding to target mRNA exhibited highly negative ΔG values for binding to target sequences, thus indicating thermodynamically feasible target binding.

The designing of therapeutic siRNA candidates involves the analysis of heat stability of the siRNA for the evaluation of its in vivo stability and functional efficiency [87]. In our study, all the 19 siRNA candidates analyzed for heat stability were found to exhibit melting temperatures considerably greater than the physiological temperature, thus indicating structural integrity in the host system. Consequently, assessment of target accessibility of the siRNA candidates indicates the efficiency of RISC-mediated endonucleolytic cleavage, which is the final step in the biological mechanism of gene silencing by siRNA [88]. siRNA 5, siRNA 49, and siRNA 50 exhibited the best scores for assessment of target accessibility; i.e., the least negative “Break targ. ΔG” values.

The immune system is armed with the required machinery to recognize foreign RNA sequences, resulting in the mediation of activation of pattern-recognition regions (PRR) for the clearance of the exogenous components [89]. Thereby, the immunogenicity of an RNA sequence in the case of siRNA-based therapy may lead to immunotoxicity [90]. Our assessment of immunogenicity of the candidate siRNAs showed that a total of 12 siRNAs were non-immunogenic (Table 3) and thus were considered for tertiary structure validation. Owing to the small number of siRNA molecules being evaluated by X-ray crystallography, NMR spectroscopy, and cryoelectronic microscopy (cryo-EM), tertiary structure prediction of siRNA is integral in understanding the respective structure-function relationship [91]. Among the 12 siRNA candidates that were analyzed for tertiary structure validation, only three siRNA candidates (siRNA 9, siRNA 15, and siRNA 17) satisfied all threshold scores for nucleic acid chemistry parameters such as: RNA sugar puckers, RNA backbone conformations, and bond angles (Table 4).

Therefore, the siRNA candidates that were found to have an acceptable tertiary structure are intended to halt the translation of distinct conserved virulence genes of *Shigella*. siRNA 9 is intended to bind to the mRNA of Ipg D gene, leading to a halt in the expression of the IpgD protein that is involved in the entry of the bacteria through ruffling of the host membrane [34]. Thus, siRNA 9 is expected to protect the host system through the maintenance of host membrane integrity. siRNA 15 and siRNA 17 are intended to target the expression of the OspB gene, thereby halting the process of host inflammatory response and resultant tissue damage [34].

## 5. Limitations of the Study

Despite the long-term prospects of the findings intended for the development of a novel therapeutic strategy for shigellosis, our study has several limitations. First of all, our analysis is solely based on the sequences of the isolates reported in the NCBI database and may not necessarily be applicable for isolates with novel mutations in these conserved virulence genes. Moreover, the implementation of siRNA in regular therapeutic applications is a distant prospect and requires extensive wet-lab validation regarding the mode of delivery into the host system. Additionally, modifications in the siRNA sequences may be required for enhanced efficacy in the host system. In addition, the prospects of the candidate siRNAs exhibiting off-target silencing or restricted functional efficiency may be pertinent due to the abundance of a myriad of genetic polymorphisms between individuals of different ethnicities. In our study, the aspect of different types of genetic polymorphisms between different ethnicities was not considered or evaluated. Although we found three siRNAs with a valid tertiary structure, we were limited to hypothesizing about their structural and functional attributes in the physiological environment, whereby extensive wet-lab studies would be required to validate such propositions. Finally, it is not expected that such novel therapeutic strategies will be of immediate clinical use, due to an elongated time span and the exorbitant cost involved in the development of such therapeutics.

## 6. Conclusions

In this study, we proposed three distinct siRNA candidates that were found to have substantial functional efficacy and non-immunogenicity with acceptable tertiary structures, which are integral for the effectiveness of siRNA in the host system. Our study provides insights into the development of a new form of therapeutics against shigellosis in the form of siRNA. Such future therapeutic strategies may have promising implications in combating the rapid development of anti-microbial resistance among *Shigella* sp. and the emergence of multi-drug resistant isolates.

## Figures and Tables

**Figure 1 molecules-27-01936-f001:**
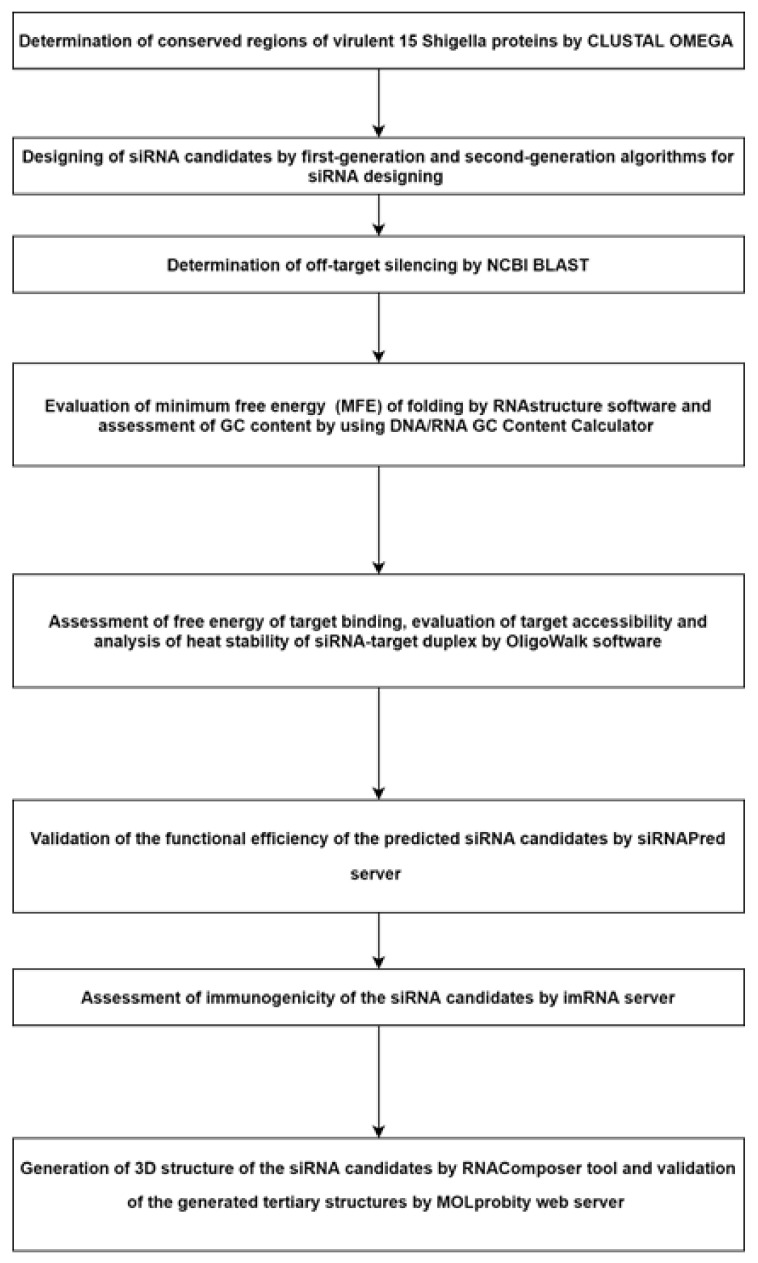
Overview of the workflow of the methodology followed in the study.

**Figure 2 molecules-27-01936-f002:**
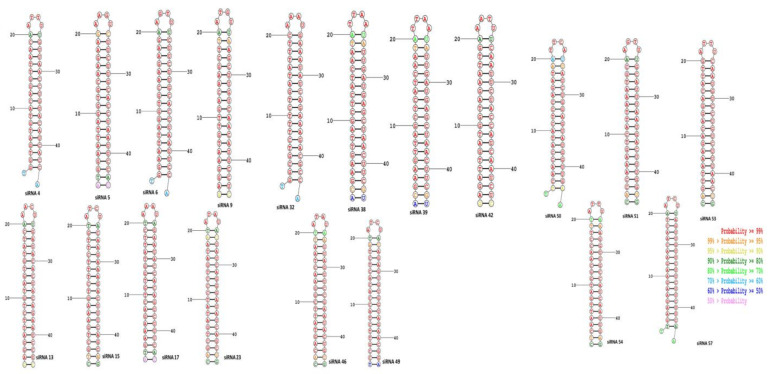
Secondary duplex structures between the siRNA candidates and their respective mRNA target sequences.

**Figure 3 molecules-27-01936-f003:**
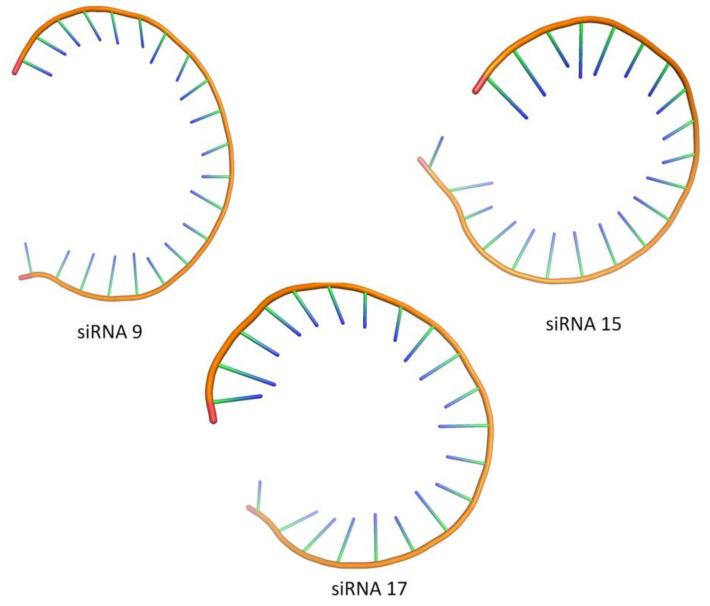
Tertiary structures of siRNA 9, siRNA 15, and siRNA 17. The validity of the tertiary structures of these siRNAs had been classified as “acceptable” on the basis of the analyzed criteria for nucleic acid geometry.

**Table 1 molecules-27-01936-t001:** List of webservers used in this study and their corresponding functions. (URL accessed on date 12 January 2021).

Name of Webserver	URL	Function
NCBI Nucleotide	https://www.ncbi.nlm.nih.gov/nucleotide	Databases consisting of nucleotide sequences from sources such as: GenBank, RefSeq, TPA, and PDB.
Clustal Omega	https://www.ebi.ac.uk/Tools/msa/clustalo/	Multiple sequence-alignment tools that use seeded guide trees and Hidden Markov Models (HMM) to generate alignments between three or more sequences.
siDirect 2.0	http://sidirect2.rnai.jp/	Candidate siRNA designing on the basis of the first generation algorithms for siRNA designing (Ui-Tei, Amarzguioui, and Reynolds rules) and using target mRNA sequences as query.
i-SCORE Designer	https://www.med.nagoya-u.ac.jp/neurogenetics/i_Score/i_score.html	Validation of potential siRNA candidates on the basis of the second generation rules for siRNA designing (i-Score‚ s-Biopredsi‚ and DSIR)
BLAST	http://www.ncbi.nlm.nih.gov/blast	Locating regions of similarity between biological sequences for the determination of off-target sequence similarity between candidate siRNA and other nucleotide sequences in the host system.
RNAstructure	https://rna.urmc.rochester.edu/RNAstructureWeb/	Prediction and validation of the secondary structures of the siRNA candidates.
OligoCalC: Oligonucleotide Properties Calculator	http://biotools.nubic.northwestern.edu/OligoCalc.html	Calculating the GC content in the predicted siRNA candidates.
OligoWalk	http://rna.urmc.rochester.edu/cgi-bin/server_exe/oligowalk/oligowalk_form.cgi	Assessment of thermodynamic stability of the predicted candidate siRNA and determination of its strand, functional efficiency, and target accessibility.
siRNAPred	http://crdd.osdd.net/raghava/sirnapred/index.html	Validation of the functional efficiency of the predicted siRNA candidates against the Main21 dataset.
imRNA	https://webs.iiitd.edu.in/raghava/imrna/sirna.php	Prediction of the immunotoxicity of the designed siRNA candidates.
RNAComposer	http://rnacomposer.cs.put.poznan.pl/	Prediction of tertiary structure siRNA candidates.
MOLprobity	http://molprobity.biochem.duke.edu/	Validation of the tertiary structure of siRNA candidates.

**Table 2 molecules-27-01936-t002:** Effective siRNA candidates for each of the conserved virulence-associated *Shigella* genes as analyzed by both the first-generation and second-generation algorithms for designing siRNAs along with their respective free energy of folding and GC content.

siRNA	Gene	Target Sequence	Predicted siRNA Duplex siRNA Candidate at 37 °C (5′ to 3′)	Combined Second-Generation Algorithm Scoring ^a^	Free Energy of Self-Folding of Guide Strand (kcal/mol)	GC Content (%)
siRNA 1	IcsA	TGGAATGGATGCGTGGTATATAA	AUAUACCACGCAUCCAUUCCA	<90% threshold	-----	-----
			GAAUGGAUGCGUGGUAUAUAA			
siRNA 2	IcsA	TACGGATTAACTCTGATATTATG	UUAUAUAUCCGCUAUUAGCAA	>90% threshold	1.0	28.57
			GCUAAUAGCGGAUAUAUAAUU			
siRNA 3	IcsA	TTGCTAATAGCGGATATATAATT	UUAUAUAUCCGCUAUUAGCAA	>90% threshold	−1.30	-----
			GCUAAUAGCGGAUAUAUAAUU			
siRNA 4	IcsA	TGGTGATGTACAGGTTAACAATT	UUGUUAACCUGUACAUCACCA	>90% threshold	1.50	38.10
			GUGAUGUACAGGUUAACAAUU			
siRNA 5	IpaJ	ATGGAATTAGGTGCAAGAAGAAG	UCUUCUUGCACCUAAUUCCAU	>90% threshold	1.80	38.10
			GGAAUUAGGUGCAAGAAGAAG			
siRNA6	IpaJ	TGGAATTAGGTGCAAGAAGAAGT	UUCUUCUUGCACCUAAUUCCA	>90% threshold	1.80	38.10
			GAAUUAGGUGCAAGAAGAAGU			
siRNA7	IpgB	AAGGATCTTACAAATCTAGTATC	UACUAGAUUUGUAAGAUCCUU	>90% threshold	1.70	28.57
			GGAUCUUACAAAUCUAGUAUC			
siRNA8	IpgD	AGCCATGATGGAACGATTAGATA	UCUAAUCGUUCCAUCAUGGCU	<90% threshold	-----	-----
			CCAUGAUGGAACGAUUAGAUA			
siRNA 9	IpgD	GAGGAATGTTGACAAGCTTAATG	UUAAGCUUGUCAACAUUCCUC	>90% threshold	1.80	38.10
			GGAAUGUUGACAAGCUUAAUG			
siRNA 10	IpgD	TGGAAATCCAAGAGATGAATACT	UAUUCAUCUCUUGGAUUUCCA	<90% threshold	-----	-----
			GAAAUCCAAGAGAUGAAUACU			
siRNA 11	MxiC	TTCAGCTATACAGGCTAAATTAT	AAUUUAGCCUGUAUAGCUGAA	<90% threshold	-----	-----
			CAGCUAUACAGGCUAAAUUAU			
siRNA 12	MxiC	GTGAAAGTGAGCAAATTCTTACT	UAAGAAUUUGCUCACUUUCAC	<90% threshold	-----	-----
			GAAAGUGAGCAAAUUCUUACU			
siRNA 13	MxiC	GAGGATTCTGTAGTGTATCAAAC	UUGAUACACUACAGAAUCCUC	>90% threshold	1.70	38.10
			GGAUUCUGUAGUGUAUCAAAC			
siRNA 14	OspB	ATGTACAAACAATCATTTCAAGA	UUGAAAUGAUUGUUUGUACAU	>90% threshold	1.70	23.80
			GUACAAACAAUCAUUUCAAGA			
siRNA 15	OspB	CTGCTGAAAGTCTTTCTTGTATC	UACAAGAAAGACUUUCAGCAG	>90% threshold	1.70	38.10
			GCUGAAAGUCUUUCUUGUAUC			
siRNA 16	OspB	ATCTTTGCTAGAGCAGATAAAAA	UUUAUCUGCUCUAGCAAAGAU	>90% threshold	−1.40	-----
			CUUUGCUAGAGCAGAUAAAAA			
siRNA 17	OspB	ATGAAAGACTGTGGTATTCTAAA	UAGAAUACCACAGUCUUUCAU	>90% threshold	1.60	33.33
			GAAAGACUGUGGUAUUCUAAA			
siRNA 18	OspF	ATGCTTTCTGCGAATGAAAGATT	UCUUUCAUUCGCAGAAAGCAU	<90% threshold	-----	-----
			GCUUUCUGCGAAUGAAAGA			
siRNA 19	OspF	TGGAAGATAACTGATATGAATCG	AUUCAUAUCAGUUAUCUUCCA	>90% threshold	1.80	28.57
			GAAGAUAACUGAUAUGAAUCG			
siRNA 20	OspF	ATGGAAGATAACTGATATGAATC	UUCAUAUCAGUUAUCUUCCAU	>90% threshold	1.80	28.57
			GGAAGAUAACUGAUAUGAAUC			
siRNA 21	OspF	TCGCAATATAGTGCTTTATTACT	UAAUAAAGCACUAUAUUGCGA	>90% threshold	−2.0	-----
			GCAAUAUAGUGCUUUAUUACU			
siRNA 22	OspG	GCCCATTCTCGGTAAGTTAATAG	AUUAACUUACCGAGAAUGGGC	<90% threshold	-----	-----
			CCAUUCUCGGUAAGUUAAUAG			
siRNA 23	OspG	CAGCTGATATCCCTGATAATATA	UAUUAUCAGGGAUAUCAGCUG	>90% threshold	1.50	38.10
			GCUGAUAUCCCUGAUAAUAUA			
siRNA 24	OspG	ATCTACAGTTGATATGTAAATTG	AUUUACAUAUCAACUGUAGAU	<90% threshold	-----	-----
			CUACAGUUGAUAUGUAAAUUG			
siRNA 25	OspG	ATCCATTACGATCTTAATACAGG	UGUAUUAAGAUCGUAAUGGAU	>90% threshold	1.60	28.57
			CCAUUACGAUCUUAAUACAGG			
siRNA 26	OspG	CGCAATATTTATGCTGAATATTA	AUAUUCAGCAUAAAUAUUGCG	>90% threshold	−2.40	-----
			CAAUAUUUAUGCUGAAUAUUA			
siRNA 27	Spa33	GACAATCAATGAACTAAAAATGT	AUUUUUAGUUCAUUGAUUGUC	<90% threshold	-----	-----
			CAAUCAAUGAACUAAAAAUGU			
siRNA 28	Spa33	AACTAAAAATGTATGTAGAAAAC	UUUCUACAUACAUUUUUAGUU	>90% threshold	1.60	19.05
			CUAAAAAUGUAUGUAGAAAAC			
siRNA 29	Spa33	ATGTATGTAGAAAACGAATTATT	UAAUUCGUUUUCUACAUACAU	>90% threshold	1.60	28.10
			GUAUGUAGAAAACGAAUUAUU			
siRNA 30	Spa33	TTCAAGTTTCCCGATGACATAGT	UAUGUCAUCGGGAAACUUGAA	>90% threshold	−1.30	-----
			CAAGUUUCCCGAUGACAUAGU			
siRNA 31	VirF	TTCAACAAATCCTTCTTGATATT	UAUCAAGAAGGAUUUGUUGAA	<90% threshold	-----	-----
			CAACAAAUCCUUCUUGAUAUU			
siRNA 32	VirF	TGGCGTCTTTCTGATATTTCAAA	UGAAAUAUCAGAAAGACGCCA	>90% threshold	1.80	38.10
			GCGUCUUUCUGAUAUUUCAAA			
siRNA 33	VirF	GTCTTTCTGATATTTCAAATAAC	UAUUUGAAAUAUCAGAAAGAC	>90% threshold	−1.40	-----
			CUUUCUGAUAUUUCAAAUAAC			
siRNA 34	VirF	AACTTGAATTTATCAGAAATAGC	UAUUUCUGAUAAAUUCAAGUU	<90% threshold	-----	-----
			CUUGAAUUUAUCAGAAAUAGC			
siRNA 35	VirA	GCCTGAACAACGAGTTATTAACA	UUAAUAACUCGUUGUUCAGGC	<90% threshold	-----	-----
			CUGAACAACGAGUUAUUAACA			
siRNA 36	VirA	CTGAACAACGAGTTATTAACAAT	UGUUAAUAACUCGUUGUUCAG	<90% threshold	-----	-----
			GAACAACGAGUUAUUAACAAU			
siRNA 37	VirA	TACGAAGTTAGCTCATCAATATT	UAUUGAUGAGCUAACUUCGUA	>90% threshold	−0.90	-----
			CGAAGUUAGCUCAUCAAUAUU			
siRNA 38	VirA	ACGAAGTTAGCTCATCAATATTA	AUAUUGAUGAGCUAACUUCGU	>90% threshold	1.60	33.33
			GAAGUUAGCUCAUCAAUAUUA			
siRNA 39	VirA	CTCCAGAAAGTCGTCAAGTATCA	AUACUUGACGACUUUCUGGAG	>90% threshold	1.60	42.86
			CCAGAAAGUCGUCAAGUAUCA			
siRNA 40	VirB	CTCCATTCTGGTAATAAAGTTTC	AACUUUAUUACCAGAAUGGAG	<90% threshold	-----	-----
			CCAUUCUGGUAAUAAAGUUUC			
siRNA 41	VirB	AACGAATGTACGCGATCAAGAAT	UCUUGAUCGCGUACAUUCGUU	<90% threshold	-----	-----
			CGAAUGUACGCGAUCAAGAAU			
siRNA 42	VirB	GAGATTGATGGTAGAATTGAAAT	UUCAAUUCUACCAUCAAUCUC	>90% threshold	1.80	33.33
			GAUUGAUGGUAGAAUUGAAAU			
siRNA 43	VirB	AACTAGCAAACGATATACAAACA	UUUGUAUAUCGUUUGCUAGUU	>90% threshold	1.80	28.57
			CUAGCAAACGAUAUACAAACA			
siRNA 44	VirB	TAGTTCTACACTACCAATATTAA	AAUAUUGGUAGUGUAGAACUA	>90% threshold	−1.10	-----
			GUUCUACACUACCAAUAUUAA			
siRNA 45	IpaA	GGGAAAGAAGATGTGTTAAGAAG	UCUUAACACAUCUUCUUUCCC	<90% threshold	-----	-----
			GAAAGAAGAUGUGUUAAGAAG			
siRNA 46	IpaA	CACAGTATTCGGAACTAATTATA	UAAUUAGUUCCGAAUACUGUG	>90% threshold	1.90	30.96
			CAGUAUUCGGAACUAAUUAUA			
siRNA 47	IpaA	TTGCACTATAGCACAACAACACA	UGUUGUUGUGCUAUAGUGCAA	<90% threshold	-----	-----
			GCACUAUAGCACAACAACACA			
siRNA 48	IpaA	CTCCTCAATACTGAAGTATCATC	UGAUACUUCAGUAUUGAGGAG	>90% threshold	−3.20	-----
			CCUCAAUACUGAAGUAUCAUC			
siRNA 49	IpaA	TCCGTTCTACCACACTCTATATC	UAUAGAGUGUGGUAGAACGGA	>90% threshold	1.70	42.86
			CGUUCUACCACACUCUAUAUC			
siRNA 50	IpaA	TTCAACCATACTCCAGATAATTC	AUUAUCUGGAGUAUGGUUGAA	>90% threshold	1.50	35.71
			CAACCAUACUCCAGAUAAUUC			
siRNA 51	IpaB	CACCAAAGTCATTAAATGCAAGT	UUGCAUUUAAUGACUUUGGUG	>90% threshold	1.50	33.33
			CCAAAGUCAUUAAAUGCAAGU			
siRNA 52	IpaB	AAGAAATACAACTCACTATCAAA	UGAUAGUGAGUUGUAUUUCUU	>90% threshold	1.50	28.57
			GAAAUACAACUCACUAUCAAA			
siRNA 53	IpaB	CAGTTAAAGACAGGACATTGATT	UCAAUGUCCUGUCUUUAACUG	>90% threshold	1.80	35.71
			GUUAAAGACAGGACAUUGAUU			
siRNA 54	IpaB	CTCAATTGATGGCAACCTTTATT	UAAAGGUUGCCAUCAAUUGAG	>90% threshold	1.40	35.71
			CAAUUGAUGGCAACCUUUAUU			
siRNA 55	IpaB	CTCCTTTCAGATGCATTTACAAA	UGUAAAUGCAUCUGAAAGGAG	<90% threshold	-----	-----
			CCUUUCAGAUGCAUUUACAAA			
siRNA 56	IpaB	GGCCAATTGCAGGAAGTAATTGC	AAUUACUUCCUGCAAUUGGCC	<90% threshold	-----	-----
			CCAAUUGCAGGAAGUAAUUGC			
siRNA 57	IpaC	TTGAAGAAGAAGAACAACTAATC	UUAGUUGUUCUUCUUCUUCAA	>90% threshold	1.80	30.96
			GAAGAAGAAGAACAACUAAUC			
siRNA 58	IpaC	AAGAAGAAGAACAACTAATCAGT	UGAUUAGUUGUUCUUCUUCUU	<90% threshold	1.70	30.963
			GAAGAAGAACAACUAAUCAGU			

^a^ Combined score obtained from the results of the second-generation tools for siRNA designing, namely: i-SCORE, s-Biopredsi, Katoh, and DSIR. Free energy of self-folding and evaluation of GC content.

**Table 3 molecules-27-01936-t003:** Designed siRNA molecules with their respective free energy of binding with target, melting temperature of siRNA–target duplex, target accessibility, functional efficiency, and immunogenicity.

siRNA	Target Sequence	Free Energy of Binding with Target (kcal/mol)	Melting Temperature of siRNA–Target Duplex(°C)	End-Diff ^a^ (kcal/mol)	Break-Targ.∆G ^b^ (kcal/mol)	Probability of Being Efficient siRNA	siRNA Validity Score (Binary)	Immunogenicity
siRNA 4	TGGTGATGTACAGGTTAACAATT	−32.6	79.1	1.76	−1.8	0.951	0.901	Non-immunomodulatory (IMscore:4.2)
siRNA 5	ATGGAATTAGGTGCAAGAAGAAG	−32.1	79.8	0.17	−0.1	0.951	0.95	Non-immunomodulatory (IMscore:3.5)
siRNA 6	TGGAATTAGGTGCAAGAAGAAGT	−32.5	78.2	0.17	−0.1	0.951	0.95	Non-immunomodulatory (IMscore:4.2)
siRNA 9	GAGGAATGTTGACAAGCTTAATG	−31.8	78.2	2.33	−1.4	0.961	0.967	Non-immunomodulatory (IMscore:3.4)
siRNA 13	GAGGATTCTGTAGTGTATCAAAC	−32.7	78.7	2.33	−0.7	0.974	0.975	Non-immunomodulatory (IMscore:2.6)
siRNA 15	CTGCTGAAAGTCTTTCTTGTATC	−31	78.3	2.09	−1.8	0.967	1.026	Non-immunomodulatory (IMscore:2.7)
siRNA 17	ATGAAAGACTGTGGTATTCTAAA	−30.1	78.9	0.03	−0.9	0.947	1.026	Non-immunomodulatory (IMscore:3.1)
siRNA 23	CAGCTGATATCCCTGATAATATA	−32.4	79.1	2.09	−0.9	0.96	0.989	Immunomodulatory (IMscore:4.7)
siRNA 32	TGGCGTCTTTCTGATATTTCAAA	−31.9	76.6	1.76	−1.8	0.957	1.009	Immunomodulatory (IMscore:4.8)
siRNA 38	ACGAAGTTAGCTCATCAATATTA	−29.4	74.9	1.7	−0.3	0.931	1.012	Non-immunomodulatory (IMscore:4.0)
siRNA 39	CTCCAGAAAGTCGTCAAGTATCA	−33	80.4	2.16	−0.3	0.952	0.949	Non-immunomodulatory (IMscore:2.8)
siRNA 42	GAGATTGATGGTAGAATTGAAAT	−30.2	74	1.87	−1	0.968	0.965	Immunomodulatory (IMscore:5.6)
siRNA 46	CACAGTATTCGGAACTAATTATA	−28.6	74.6	1.23	−0.3	0.936	0.971	Non-immunomodulatory (IMscore:2.2)
siRNA 49	TCCGTTCTACCACACTCTATATC	−34.3	82	1.03	−0.1	0.935	1.006	Non-immunomodulatory (IMscore:1.1)
siRNA 50	TTCAACCATACTCCAGATAATTC	−30.3	78.5	1.46	0	0.886	0.886	-
siRNA 51	CACCAAAGTCATTAAATGCAAGT	−29	76.1	0.13	−0.3	0.818	0.818	-
siRNA 53	CAGTTAAAGACAGGACATTGATT	−31.3	78.1	0.79	−1.1	0.926	0.909	Immunomodulatory (IMscore:5.6)
siRNA 54	CTCAATTGATGGCAACCTTTATT	−31	78	1.23	−1	0.891	-	-
siRNA 57	TTGAAGAAGAAGAACAACTAATC	−27.8	76.2	0.33	−0.2	0.906	0.999	Immunomodulatory (IMscore:7.3)

^a^ The free energy difference between the 5′ and 3′ end of the antisense strand of siRNA. ^b^ The free energy cost for opening base pairs in the region complementary to the target.

**Table 4 molecules-27-01936-t004:** Validation scores for the predicted 3D structures of the 11 non-immunogenic siRNA candidates.

siRNA	Nucleic Acid Geometry				All Atom Contacts (clashsccore, Percentile) ^b^	Validity of Predicted 3D Structure of siRNA
	Probability of Wrong Sugar Puckers ^a^ (%)	Bad Backbone Confirmations ^a^ (%)	Bad Bonds ^a^ (%)	Bad Angles ^a^ (%)		
siRNA 4	0.00	0.00	0.00	0.00	36.81, 9th	Low
siRNA 5	0.00	0.00	0.00	0.00	30.67, 15th	Low
siRNA 6	0.00	0.00	0.00	0.00	36.81, 9th	Low
siRNA 9	0.00	0.00	0.00	0.00	19.76, 33rd	Acceptable
siRNA 13	0.00	0.00	0.00	0.00	24.1, 23rd	Low
siRNA 15	0.00	4.76	0.00	0.00	19.23, 35th	Acceptable
siRNA 17	0.00	0.00	0.00	0.00	19.61, 34th	Acceptable
siRNA 38	4.76	57.14	0.00	0.00	22.52, 26th	Low
siRNA 39	0.00	0.00	0.00	0.00	22.46, 26th	Low
siRNA 46	0.00	0.00	0.00	0.00	29.76, 16th	Low
siRNA 49	0.00	28.57	0.00	0.00	19.03, 35th	Low

^a^ 5% threshold is allowed for acceptance of the criteria for a valid 3D structure. 100th percentile is the best among the structures for a comparable resolution; 0th percentile ranks to be the worst. ^b^ Clashscore is the parameter that represents a comparable resolution.

## Data Availability

All required data have been presented in the manuscript. Results section and in the Appendix A.

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
