# Peer review of "A Comprehensive Computational Investigation into the Conserved Virulent Proteins of *Shigella* species Unveils Potential Small-Interfering RNA Candidates as a New Therapeutic Strategy against Shigellosis"

_molecules, 2022, doi:10.3390/molecules27061936_

Round 1

Reviewer 1 Report

Paper is systematically written with appropriate methodology and results. However,

  1. Figure 2 needs improvement in resolution and clarity.
  2. Table 3 mentions 12 non-immunogenic siRNA candidates, but only 11 siRNA are tabulated. Correct the discrepancy.
  3. Make a new table listing all the name and URL of the webservers used along with a brief description of their function.
  4. Minor improvement in English language is suggested.

Author Response

Thank you for your kind review of our manuscript. Please find the point to point responses to each of your comments and suggestions in the attached file. 

Reviewer 2 Report

This article is of particular importance. But it requires improvement in following section:

  • Method section is poorly written. For example, authors are mentioning weblink instead of the original research work. All weblinks (blast, clustalo, etc.) should be referenced with the actual publication in the Methods section.
  • All siRNAs should be shared as supplementary file to confirm research reprocucibiity.
  • BLAST parameters are not clear; if you select e-value 10, many statistically non-significant hits will pop up. Authors need to select strict threshold and re-run the experiments.
  • From Table 3 it is not clear, based on which parameters of 3D structure, three siRNAs were selected. Authors need to justify the selection and threshold with appropriate reference. Any arbitrary threshold should not be considered.
  • Change the background color of Figure 3 to enhance the visibility of the siRNAs

Author Response

(The authors gave the same response as above.)

Reviewer 3 Report

The manuscript was well prepared for the publication, however, I have some issues which the authors need to clarify.

  •  Could the static structure of siRNA well reflect the acceptable structure as in fact the condition of the siRNA would be in the aqueous solution so the dynamic would be more effective to see if the structure is acceptable?
  • For the binding energy of siRNA to the target, why do the authors not include the positive control as the relative reference, as I understand the binding energy is relative and the predicted binding energy would need the reference for physical binding affinity evaluation? Please author clarify the point and the true meaning of binding energy and how to interpret this values in the physical manner apart from the binding affinity comparison. I do understand that the wet lab is still needed but at least only pure prediction could inform some convincing information.

Author Response

(The authors gave the same response as above.)

Reviewer 4 Report

I love reading this paper "A comprehensive computational investigation into the conserved virulent proteins of Shigella species unveils potential small-interfering RNA candidates as a new therapeutic strategy against shigellosis" and would like to accept it for publication. However, some minor changes are suggested prior to publication.

  1. A thorough English check is needed to remove minor grammatical mistakes.
  2.  Figures quality needs to be improved. 
  3. Discuss study limitations. 

Author Response

(The authors gave the same response as above.)

Round 2

Reviewer 1 Report

  1. Minor corrections and language improvement like following can be done throughout the manuscript: Pg 21, line 445: In our study, we discuss the aspect...
  2. Please do a plagiarism check for the entire manuscript  

Reviewer 2 Report

Authors have implemented the suggested changes.

Reviewer 3 Report

The manuscript should be accepted.